# A Systematic Review of Surface Electromyography in Sarcopenia: Muscles Involved, Signal Processing Techniques, Significant Features, and Artificial Intelligence Approaches

**DOI:** 10.3390/s25072122

**Published:** 2025-03-27

**Authors:** Alessandro Leone, Anna Maria Carluccio, Andrea Caroppo, Andrea Manni, Gabriele Rescio

**Affiliations:** National Research Council of Italy, Institute for Microelectronics and Microsystems, 73100 Lecce, Italy; andrea.caroppo@cnr.it (A.C.); andrea.manni@cnr.it (A.M.); gabriele.rescio@cnr.it (G.R.)

**Keywords:** surface electromyography, sarcopenia, wearable sensors, ageing population, muscles quality, machine learning, deep learning

## Abstract

Sarcopenia, affecting between 1–29% of the older population, is characterized by an age-related loss of skeletal muscle mass and function. Reduced muscle strength, either in terms of quantity or quality, and poor physical performance are among the criteria used to diagnose it. The current gold standard methods to evaluate sarcopenia are limited in terms of their cost, required expertise, and portability. A possible alternative for sarcopenia detection and monitoring is surface electromyography, which offers comprehensive information on muscle function, but a systematic synthesis of the existing literature is lacking. This systematic review aims to evaluate the application of sEMG in diagnosing and monitoring sarcopenia, focusing on the muscles involved, signal processing techniques, artificial intelligence models, and statistical analysis methods used for data interpretation. Following PRISMA guidelines, a search was performed in PubMed, Scopus, and IEEE databases from 2014 up to December 2024. Original studies using sEMG for sarcopenia diagnosis or assessment in older populations were included. After removing duplicates, 145 articles were identified, of which 18 were included in the final analysis. The findings indicate a growing interest in the adoption of sEMG in sarcopenia assessment. However, methodological heterogeneity among studies limits comparability. sEMG represents a promising option for the early detection of sarcopenia, but standardized guidelines for data collection and interpretation are needed. Future studies should focus on clinical validation and results reproducibility.

## 1. Introduction

Sarcopenia, characterized by a progressive and widespread loss of skeletal muscle mass and function, is associated with physical disability, a worse quality of life, and a greater mortality rate among elderly people [1]. In addition to existing risk factors, it is common in older persons as an age-related process driven by lifestyle and genetic factors that occur [2]. Nowadays, sarcopenia is commonly recognized as a unique clinical syndrome with important physiological and functional implications, despite first being thought of as a normal part of ageing. Sarcopenia is currently being investigated to improve diagnosis and treatment applying current knowledge of its pathophysiology [3]. The prevalence of sarcopenia varies widely depending on the diagnostic criteria and population studied. Using definitions consistent with the European Working Group on Sarcopenia in Older People (EWGSOP), studies have reported a prevalence of EWGSOP-defined sarcopenia of 1–29% (up to 30% in women) for older adults living in the community, 14–33% (up to 68% in men) for those living in long-term care institutions and 10% for those in acute hospital care. These variations reflect the influence of age, health status, and care environments, with higher prevalence observed in older and more medically complex individuals [4,5]. Early diagnosis of sarcopenia in community-dwelling older adults is important, and appropriate nutrition and exercise interventions are needed to reduce the dangerous consequences of sarcopenia and ultimately reduce the risk of hospitalization in elderly [6]. This condition has an estimated economic impact in the billions of dollars annually, driven by increased healthcare utilization, prolonged hospitalizations, and extensive rehabilitation needs [7]. In recent years, global guidelines for the diagnosis and management of sarcopenia have been developed, notably by the EWGSOP [8,9], the Asian Working Group for Sarcopenia (AWGS) [10,11], the International Working Group on Sarcopenia (IWGS) [12], and the Foundation of National Institutes of Health (FNIH) [13]. These guidelines establish diagnostic criteria based on three key parameters: muscle mass, muscle strength, and physical performance, aiming to standardize diagnostic practices across diverse healthcare settings. Among these, the most widely cited definition is that proposed by the EWGSOP, supported by the AWGS, which has gained endorsement from numerous international scientific societies for both clinical practice and research. According to the EWGSOP2, sarcopenia is diagnosed in individuals with low muscle strength accompanied by low muscle mass or quality [1]. This definition highlights several contributing factors, including reduced muscular strength, diminished muscle quantity or quality, and impaired physical performance [9]. The diagnostic process for sarcopenia involves a multi-step approach to assess muscle deterioration comprehensively. It evaluates physical functionality, muscle strength, muscle quantity, and muscle quality through a combination of tools and criteria recommended by clinical guidelines. In Figure 1, the algorithm to diagnose sarcopenia in clinical practice is reported.

The SARC-F questionnaire is typically used as an initial screening tool to assess self-reported physical limitations. This questionnaire evaluates five parameters: strength, assistance in walking, rising from a chair, climbing stairs, and falls [14]. A value grater than or equal to four indicates a higher risk of sarcopenia, warranting further diagnostic evaluation [15]. Muscle strength, a primary indicator of sarcopenia, is commonly assessed using handgrip strength, measured with a dynamometer. This method provides a reliable indicator of overall muscle functionality, with cutoff values established by the EWGSOP and AWGS [9,11,16]. Muscle mass is typically quantified using imaging techniques such as dual-energy X-ray absorptiometry (DXA) or bioelectrical impedance analysis (BIA). DXA offers detailed insights into body composition, while BIA is a more accessible and cost-effective alternative [17]. Advanced imaging methods, such as magnetic resonance imaging (MRI) and computed tomography (CT), allow high-resolution assessments of muscle quality, including the detection of intramuscular fat infiltration—a hallmark of sarcopenia-related muscle degeneration [17]. Physical performance is another critical parameter commonly evaluated using gait speed or the Short Physical Performance Battery (SPPB). Gait speed, measured over a defined distance, is an objective indicator of functional capacity, with slower speeds often reflecting advanced sarcopenia [17]. The SPPB, combining gait speed, balance, and chair stand tests, offers a more comprehensive evaluation. While methods such as DXA, MRI, handgrip strength and gait speed tests provide reliable assessments, they face significant barriers, including high cost, limited accessibility and the need of specialised equipment and highly qualified medical personnel. These challenges make such tools less practical for widespread use, particularly in resource-limited settings or for early diagnosis [18]. As a result, there is a growing need for alternative diagnostic approaches that are less invasive, more accessible and cost-effective, and that allow earlier and more widespread screening for sarcopenia.

To overcome these limitations, researchers have increasingly explored innovative, non-invasive technologies. Among these, surface electromyography (sEMG) has emerged as a promising alternative [19,20]. This technique records muscle electrical activity in real-time, providing unique insights into neuromuscular function, including muscle activation patterns and recruitment strategies [21]. DXA and MRI mainly assess muscle composition, whereas sEMG focuses on neuromuscular alterations, which may manifest before muscle mass loss. This makes sEMG particularly valuable for early detection of neuromuscular dysfunctions and also for long-term monitoring of disease progression and response to interventions. Recent advances in signal processing and wearable sensor technology have further improved its scalability and feasibility for assessing muscle health, making it an attractive option for both diagnosis and monitoring. Although requiring specialized personnel for electrodes placement and signal analysis, the equipment required is generally more accessible and less expensive compared to advanced imaging techniques. In addition, unlike DXA and MRI, some commercial sEMG solutions are available as portable devices that can be used at home, facilitating long-term monitoring without hospital visits. Widely used in sports and rehabilitation research [22], sEMG has proved efficient in the electrophysiological evaluation of muscle contraction and force generation. In the context of sarcopenia, its ability to detect neuromuscular impairment before significant muscle loss or functional decline becomes a valuable tool for early diagnosis. In addition, its application goes beyond diagnosis. In fact, it can be used to monitor disease progression and assess response to interventions in sarcopenic patients. The aim of this systematic review is to comprehensively evaluate the use of sEMG in sarcopenia diagnosis and monitoring. It specifically aims to find and analyze research that employed sEMG to identify neuromuscular abnormalities related to sarcopenia, including significant information about the examined muscle groups, performed motor tasks, signal processing methods, and diagnostic techniques. To guide future research and clinical implementation, this review aims to provide an organized and current perspective on the feasibility, reliability, and clinical utility of sEMG in sarcopenia assessment by synthesizing the available information. A systematic review is necessary due to the growing interest in non-invasive and simpler diagnostic techniques for sarcopenia, but the field remains fragmented with heterogeneous methodologies and inconsistent results across studies. While sEMG is widely used in rehabilitation and sports science, its application in sarcopenia diagnosis lacks a consolidated framework. A rigorous synthesis of existing research is needed to fill knowledge gaps, standardize diagnostic approaches, evaluate clinical applicability, and inform future research., The remainder of the article is organized as follows: Section 2 describes the materials and methods, including the article selection process, inclusion and exclusion criteria, and an overview of the application of sEMG in the detection of sarcopenia. Section 3 focuses on the muscles and motor tasks commonly analyzed during sEMG signal acquisition. Section 4 reviews sEMG recording techniques, including electrode placement and equipment used. Section 5 examines signal processing methods, discussing filtering, preprocessing, feature extraction, and advanced processing approaches. Systems for sarcopenia detection/assessment, covering Artificial Intelligence (AI) models and Statistical Analysis (SA) methods, are detailed in Section 6. Section 7 gives an overview of other sarcopenia-correlated fields of research. Finally, Section 8 discusses key findings, implications, and future directions for research in this area while Section 9 reports the conclusions.

## 2. Materials and Methods

An extended version of the Preferred Reporting Items for Systematic Reviews and Meta-Analyses (PRISMA) [23] was used as the methodological framework for this review. In particular, the PRISMA-S guideline [24], a 16-item checklist, was adopted to comprehensively address various aspects of the systematic review search process. PRISMA-S builds on the original PRISMA framework and its extensions, providing a structured checklist to assist interdisciplinary researchers and reviewers ensuring all critical components of a systematic review are accurately reported and reproducible. The literature search was conducted using the Scopus, PubMed, and IEEE databases to identify relevant studies published between 2014 and December 2024.

The search strategy involved a comprehensive combination of keywords and related terms concerning sEMG, sarcopenia detection, and clinical assessment of muscle characteristics related to sarcopenia (i.e., muscle mass loss and muscle strength). The search terms were selected based on the main research question, “How is surface electromyography used to detect/diagnose or assess sarcopenia?”. Given the novelty of the topic and the limited number of specific studies, the query was broadened to include all terms related to sarcopenia, along with diagnostic parameters for sarcopenia, to capture as many relevant studies as possible. The following subsections outline the inclusion and exclusion criteria used for the final screening of the analyzed paper. In addition, an overview of current research trends in the detection of sarcopenia with sEMG and the identification of sarcopenia-related parameters using sEMG analysis is provided.

### 2.1. Article Selection, Inclusion and Exclusion Criteria

The queries shown in Table 1 returned a total of 198 articles, 137 from Scopus, 60 from PubMed, and 1 from IEEE. A manual search was also performed to ensure comprehensive coverage. Only articles published from January 2014 to December 2024 were included. During the screening phase, 53 duplicates were removed, leaving 145 articles, which were then reviewed based on their titles and abstracts, and full-text availability was also verified. PDF copies of all remaining articles were then downloaded.

The eligibility criteria for inclusion in the review were as follows:-Peer-reviewed articles published in indexed journals.-Conference papers, included due to their contribution to emerging research in the field.-Studies using sEMG for analysis, detection or assessment of sarcopenia- related parameters.-Studies conducted on human participants.

On the other hand, the eligibility criteria for exclusion in the review were as follows:-Articles published in languages other than English.-Articles for which full text was not available.-Studies that did not mention sarcopenia, related parameters, or sEMG.-Studies primarily evaluating nutritional interventions or focusing on unrelated conditions-Studies involving non-skeletal muscles.

After applying the inclusion and exclusion criteria, 64 articles were identified. To ensure a structured analysis, these studies were categorized into three groups based on their focus: “Diagnosis/Detection”, “Neuromuscular Function”, and “Ageing Muscles”. This categorization highlights different research trends and supports a comprehensive exploration of the field. The first category includes studies that are perfectly on target. They focus on the use of sEMG signals for the assessment, diagnosis, and monitoring of sarcopenia and related musculoskeletal conditions. These studies often integrate advanced techniques such as Machine Learning (ML), wearable technologies, and digital biomarkers to develop innovative and portable solutions. Many are designed for home use and aim to improve care for the elderly or patients with muscular frailty. The overall goal is to develop practical and scientifically robust tools for the early diagnosis, monitoring, and classification of sarcopenia, promoting an integrated and accessible technological approach.

The second group includes studies that focus on the diagnosis, assessment, and improvement of neuromuscular function in the elderly and individuals with age-related musculoskeletal conditions. These studies examine the role of neuromuscular characteristics (e.g., motor unit recruitment, firing, muscle mass) in strength, motor control, and fatigue using sEMG or wearable devices; the effects of targeted interventions (e.g., exercise training) on motor function and quality of life; the relationship between muscle structure (e.g., mass, intramuscular fat) and physical performance using predictive or comparative methods. The overall goal is to develop strategies to prevent, diagnose, and treat sarcopenia and simultaneously improve physical performance and quality of life in older adults. Studies in the third category focus on age-related neuromuscular changes and how they affect strength, motor control, fatigue, and muscle performance. They examine changes in motor units, muscle fibers, and mechanical properties, comparing young and older adults to better understand the mechanisms behind loss of strength and mobility. In addition, they report the development of tools and strategies to monitor and improve muscle function in older adults to promote quality of life and prevent functional decline. These categories and the topics discussed in these studies have been included because they provide valuable insights into current research trends in the field. After reviewing the full text of these articles, studies including patients with sarcopenia or pre-sarcopenia as participants and studies related to the detection/diagnosis or evaluation of sarcopenia were included. The study selection was conducted by two independent reviewers, with discrepancies resolved through consensus or consultation with a third reviewer. The PRISMA flowchart summarizing the selection process is provided in Figure 2, which illustrates each step from initial database searches to the final inclusion of 18 studies for detailed analysis. As highlighted in the inclusion criteria list, a number of conference papers [25,26,27] were also included among the selected studies, providing relevant insights into emerging methodologies and applications of sEMG in sarcopenia research.

### 2.2. Overview on sEMG in Sarcopenia Detection

sEMG has emerged as a promising non-invasive tool for the assessment of sarcopenia, offering unique advantages over traditional diagnostic methods. The origins of sEMG date back to the early 20th century, with initial applications focused on assessing basic muscle function and motor control. Pioneers such as Adrian et al. [28] conducted early work on detecting the electrical activity of muscle fibers, setting the stage for the development of sEMG as a field. In the decades that followed, sEMG technology progressed rapidly, with advances in electrode design and signal processing allowing for more detailed analysis of muscle function in dynamic settings [29]. By the 1970s, sEMG had become a popular tool in clinical and athletic settings to assess muscle activation, fatigue, and motor control [30]. Today, it is widely used in research and clinical practice for the diagnosis of neuromuscular diseases and the assessment of rehabilitation progress. This evolution highlights the increasing sophistication and adaptability of sEMG in the study of muscle health across populations and contexts. Compared to imaging techniques such as DXA and MRI, or functional assessments such as handgrip strength testing, sEMG offers several distinct advantages:-Non-invasive and Real-Time Monitoring: sEMG enables non-invasive, real-time analysis of muscle activity during both static and dynamic tasks, offering insights into natural movement conditions [21].-Detection of Neuromuscular Impairments: Unlike traditional methods, sEMG captures neuromuscular activity, including recruitment patterns, activation timing, and motor unit firing rates. These changes often precede visible muscle loss in sarcopenia [31,32].-Assessment of Fatigue and Endurance: Parameters like the median frequency and root mean square amplitude help assess muscle fatigue and endurance, key factors in sarcopenia-related functional decline [33,34].

The unique ability of sEMG to capture these aspects of muscle function can complement traditional diagnostic techniques to provide a comprehensive view of muscle health in individuals with or at risk of sarcopenia. One of the most promising applications of sEMG in the assessment of sarcopenia is its potential role in early detection. Studies have shown that neuromuscular changes detectable by sEMG, such as altered recruitment patterns and reduced motor unit firing rates, can occur before significant muscle atrophy or functional decline [35,36]. Early detection of these changes could allow clinicians to initiate preventive interventions earlier, potentially slowing the progression of sarcopenia [37]. In addition, recent advances in wearable sEMG devices allow for continuous monitoring, making it possible to track muscle function longitudinally. This capability is particularly useful for high-risk populations, including elderly people or those with metabolic conditions that predispose them to sarcopenia. Continuous sEMG monitoring can detect a gradual decline in neuromuscular function, providing a “wake-up call” that traditional tests may not detect in the early stages of sarcopenia. Studies specifically focused on the diagnosis, detection, or assessment of sarcopenia using sEMG signals are limited as they represent an emerging research area. However, the analysis of muscle activity to identify parameters associated with sarcopenia has a longer history of investigation. In addition, there are other categories of studies that address conditions related to sarcopenia, such as sarcopenic dysphagia. Although these studies were excluded from the main scope of this review, they are briefly discussed in a separate section due to their complementary insights into the broader implications of this condition. On the other hand, Figure 3 illustrates the year-by-year distribution of the 64 full-text articles reviewed as they relate to sarcopenia, sarcopenia parameters, and sEMG. The articles’ distribution shows a clear trend in research activity, with a noticeable increase from 2020, reaching its peak in 2024. This indicates growing scientific interest in this topic, likely driven by advances in technologies such as sEMG, ML, and wearable devices, as well as increasing awareness of sarcopenia as a significant health issue. The limited number of studies between 2014 and 2019 suggests that this research area was initially underexplored. However, the strong increase after 2020 may reflect the emergence of innovative tools and methodologies, along with a greater focus on age-related conditions. This trend suggests a shift from a niche area of study to a more established and expanding field of research, with promising implications for early sarcopenia diagnosis, monitoring, and treatment.

## 3. Muscles and Motor Tasks for sEMG Signal Acquisition

As mentioned in the introduction, the identification of sarcopenia using sEMG analysis is an emerging research area. There is currently no established gold standard for which muscles to target or which motor tasks to use during signal acquisition. Recent investigations have examined various body regions and motor tasks, with a predominant focus on the lower limbs due to their central role in mobility and the physical performance metrics often associated with sarcopenia diagnosis. To comprehensively assess sarcopenic conditions, this review includes studies analyzing the correlation between electromyographic signals, muscle strength, muscle quality, and postural balance in sarcopenic individuals.

### 3.1. Commonly Analyzed Muscles

The selection of muscles in sEMG studies varies depending on their relevance to physical performance and sarcopenia-related changes in muscle function. The lower limbs are often examined for their critical role in mobility and balance, while some studies focus on the upper limbs and core muscles to comprehensively assess neuromuscular impairment.

#### 3.1.1. Lower Limb Muscles

Several studies have focused on specific muscles in the lower extremities. In Hirono et al. [36], the difference in motor unit firing patterns between community-dwelling elderly individuals with normal and low skeletal muscle mass was investigated. In particular, the Vastus Lateralis (VL) was studied to investigate neuromuscular fatigue and recruitment patterns. Leone et al. [38] focused on the Gastrocnemius Lateralis (GL) and Tibialis Anterior (TA) muscles developing a hardware/software platform to classify the severity of sarcopenia using sEMG signals. In Piasecki et al. [37], to confirm the remodeling of the motor unit in sarcopenic subjects, the VL and TA muscles were investigated. In Hung et al. [25], a wearable device integrating sEMG and G-sensors was used to develop an ML-based classification system for sarcopenia, focusing on lower limb muscles. In Kumar et al. [39], the Rectus Femoris (RF), Biceps Femoris (BF), TA, and GL were analyzed and a risk classification system for sarcopenia was proposed. In Zhang et al. [40] to study postural balance disturbances in elderly people with sarcopenia using sEMG, the evaluated muscles involved the Gluteus Maximus (GM), RF, BF, TA, and GL. RF was also targeted in Imrani et al. [41] where high-density sEMG (HD-sEMG) was used to monitor muscle ageing and inactivity-related changes. Finally, Godoy et al. [26] focused on the RF and BF, evaluating sEMG features as biomarkers for diagnosis and monitoring of sarcopenia.

#### 3.1.2. Upper Limb Muscles

Upper limb muscles have been studied for their potential in the diagnosis of sarcopenia, particularly regarding muscle function and grip strength. In Li et al. [42], the authors analyzed the Brachioradialis (BRA), Flexor Carpi Radialis (FCR), Flexor Digitorum Superficialis (FDS), Flexor Carpi Ulnaris (FCU), Extensor Carpi Ulnaris (ECU), and Extensor Digitorum (ED), and proposed an ML-based screening method. In Jin et al. [43], the target muscle was the Biceps Brachii to develop a portable, user-friendly sEMG acquisition system for home sarcopenia diagnosis. In He et al. [44], the authors studied forearm muscles, including the BRA and FDS, to identify sEMG patterns as biomarkers for early sarcopenia screening in community settings. In Sung et al. [45], the feasibility of using sEMG parameters to estimate muscle mass of Biceps Brachii (BB) and Triceps Brachii (TB), as well as BF and RF, was evaluated, including both lower and upper limbs.

#### 3.1.3. Core and Back Muscles

Although less commonly studied, core and back muscles are sometimes included to assess postural control and neuromuscular adaptations in sarcopenic individuals. Ma et al. [27] studied lumbar muscles, including the Multifidus (L5) and Iliocostalis Lumborum (L1), using ML to identify characteristic sEMG patterns for the sarcopenia diagnosis. A detailed summary of the analyzed studies, including the examined muscles, is presented in Table 2.

### 3.2. Motor Tasks

In the context of sEMG studies for sarcopenia diagnosis, motor tasks play a central role in the assessment of muscle performance, neuromuscular activity, and physical abilities. These tasks are divided into isometric exercises, which focus on static contraction, and dynamic exercises, which capture movement and functional performance. Isometric tasks, such as leg extension or arm flexion, involve muscle contraction without joint movement, making them ideal to assess maximum voluntary contraction (MVC) and monitoring muscle activation levels. On the other hand, dynamic tasks, which involve actions such as getting up from a chair or walking, are used to assess overall muscle functionality and motor coordination. Using sEMG to analyze both types of contractions allows for a more comprehensive understanding of muscle health. Dynamic tasks help measure endurance and functional capacity, while isometric tasks provide baseline strength data, which is essential for early sarcopenia detection.

#### 3.2.1. Isometric Exercises

Isometric tasks involve holding a position, such as maintaining ankle plantarflexion or dorsiflexion against resistance for a specified time period. This static contraction is critical for quantifying muscle strength via sEMG. Some sarcopenia-related studies do not record the sEMG signal during motor tasks. Isometric exercises are commonly used to measure parameters such as MVC and sustained voluntary contraction (SVC), providing insight into muscle activation levels. Several studies have included isometric tasks in their protocols. In Hirono et al. [36] the participants performed MVC of knee extensors twice. Jin et al. [43] used elbow flexion and extension tasks to assess MVC and SVC with a 2 kg weight. Nali et al. [42] used handgrip contractions at 20% and 50% MVC during sEMG recordings to assess forearm muscle activation. In Sung et al. [45], the authors recorded sEMG signals during MVC during Elbow Flexion (EF), Elbow Extension(EE), Knee Flexion (KF), and Knee Extension (KE). Kumar et al. [39] incorporated isometric exercises to analyze the RF, BF, and other lower limb muscles during standardized MVC protocols.

#### 3.2.2. Dynamic Exercises

On the dynamic side, the Sit-to-Stand test or gait analysis are used to monitor sarcopenia progression. These tests assess how well muscles perform during movement, which mirrors everyday activities. Dynamic exercises involving movement are often used to assess functional ability, muscle coordination, and overall physical performance. These tasks provide insight into the functional impact of sarcopenia on daily activities. Studies using dynamic exercises are reported below. Leone et al. [38] used tasks such as the Sit-to-Stand test and the gait speed test (5 m). Hirono et al. [36] used gait analysis and dynamic fatigue protocols to evaluate muscle endurance. In Ma et al. [27], forward and lateral flexion were employed as part of their dynamic task protocol to assess lumbar neuromuscular adaptation. Imrani et al. [41] incorporated activities such as walking and gait speed assessments to estimate functional performance in the lower limbs. Sung et al. [45] evaluated dynamic movements, such as standard walking and sit-to-stand transitions, to estimate muscle mass and strength using sEMG combined with other techniques. While isometric tasks focus on static muscle contraction and activation, dynamic tasks aim to assess movement-related functionality. Both types of exercises provide unique insights into sarcopenia detection and monitoring. The complete list of analyzed motor tasks is summarized in Table 2.

### 3.3. Muscles and Motor Tasks Used in Other Studies

Several studies have investigated muscle function in sarcopenic or pre-sarcopenic individuals, focusing on specific muscles and motor tasks to gain insight into neuromuscular adaptations and functional impairments. Song et al. [46] studied the RF, proposing an sEMG-based system with electrical stimulation and wearable devices for at-home muscle function evaluation. Kienbacher et al. [47] assessed the MUL, Longissimus, and IL, proposing standardized methodologies to monitor fatigue and age-related changes in back muscles, analyzing movements such as forward flexion and lateral flexion to assess postural control and lumbar fatigue. TA was analyzed during isometric voluntary contractions, fatigue-inducing tasks, and dynamic motor tasks to elucidate its role in muscle endurance and balance [48]. Ebenbichler et al. [34] explored whether fatigue indices could differentiate age-specific muscle functions in the lumbar region of individuals with chronic low back pain. Ebenbichler et al. [49] further investigated gender and age differences in neuromuscular function, focusing on the same back muscles to achieve the preclinical detection of back muscle aging, and possibly predict back muscle sarcopenia development.

The analysis of muscle activity during specific motor tasks is crucial for understanding neuromuscular dynamics and evaluating the effectiveness of rehabilitation interventions or functional assessments. In the following section, the details of sEMG recording techniques are discussed, focusing on electrode placement strategies and the different devices used for data acquisition. These aspects are critical to ensure accurate and reliable signal collection, essential to analyze the muscles involved in the motor tasks described in the previous section.

## 4. sEMG Recording Techniques

The quality of sEMG signals can be affected by several factors, including electrode placement, equipment setup and the type of motor task being performed. Careful attention to these variables is required to ensure high-quality recordings. Electrodes are placed along the longitudinal axis of the muscle fibers, typically on the muscle belly, to minimize interference or “crosstalk” from adjacent muscles. To achieve optimal signal quality, the skin must be properly prepared by shaving excess hair, cleaning with alcohol to remove oil and dead skin, and ensuring secure adhesion of the electrodes. The standard electrode spacing of 1–2 cm depends on the size of the target muscle. The choice of equipment has a significant impact on the reliability of sEMG recordings. Disposable Ag/AgCl electrodes are commonly used for their low impedance and ease of application, while reusable electrodes are better for long-term use but require more maintenance. Amplifiers play a key role in enhancing weak sEMG signals while reducing noise, with high-quality amplifiers offering significant gain (1000x–10,000x), high signal-to-noise ratios, and a common mode rejection ratio greater than 100 dB. Filters are also critical to improve signal clarity, with band-pass filters (10–500 Hz) commonly used to retain frequencies relevant to sEMG. Recording systems integrated with advanced software allow real-time signal visualization and analysis. Wireless systems are particularly useful for dynamic tasks, such as walking, for their portability and flexibility. In contrast, wired systems are preferred for isometric tasks due to their signal stability and reliability during stationary measurements. The type of motor task performed during sEMG recording also influences signal quality. Isometric tasks are ideal for measuring endurance and fatigue while ensuring stable electrode positioning. Dynamic tasks introduce additional challenges, such as motion artifacts, which can be mitigated using adhesive electrodes and wireless systems. By selecting appropriate tasks based on study objectives, researchers can better assess muscle function and neuromuscular activation. By following standardized protocols and using advanced recording technologies, researchers and clinicians can improve the reliability and diagnostic potential of sEMG to detect sarcopenia-related changes in muscle activation and performance. These practices are essential to advance the use of sEMG in both clinical and research settings.

### 4.1. Standards for Electrodes Placement

The quality of sEMG signals is strongly influenced by the placement, size, and inter-electrode distance (IED) of the electrodes. Traditionally, sEMG signals are acquired using a pair of single electrodes placed on the skin over the target muscle, with a differential amplifier providing the signal for user or software interpretation. However, the amplitude and spectral characteristics of the signal can vary significantly depending on the position and orientation of the electrodes along the muscle fibers [50]. Array electrodes are an advanced alternative to traditional single-pair electrodes, offering higher spatial resolution and the ability to record a wider range of neuromuscular activity. Arrays consist of multiple small electrodes that can record signals over a larger muscle area, making them particularly suitable for dynamic tasks and HD-sEMG analysis. These systems allow detailed mapping of muscle activation patterns, including identification of innervation zones (IZ) and tendon locations, providing critical information for positioning electrodes to avoid interference and improve signal quality [51]. To optimize electrode placement, studies [52,53,54] have emphasized the importance of positioning the electrodes between the IZ and the tendon, ensuring that no electrode overlaps the IZ over the full range of motion. For lower extremity muscles, landmarking techniques have been proposed to standardize the location of the IZ to improve consistency and accuracy across studies. In addition, standardized recommendations, such as those provided by sEMG for non-invasive assessment of muscles (SENIAM), include the use of 10 mm diameter Ag/AgCl electrodes placed at an IED of 20 mm to achieve reliable recordings across a range of muscles and motor tasks [55]. Dynamic tasks present additional challenges for electrode placement as muscle fiber positions and IZ locations shift with changes in joint angle. In such scenarios, flexible or array-based electrodes offer a more robust solution to maintain signal integrity. Standardized protocols and adherence to international guidelines, which often suggest placing electrodes on the muscle belly for optimal recording quality, further enhance the reliability and reproducibility of sEMG data [56].

### 4.2. sEMG Devices Used in Sarcopenia Studies

sEMG devices play a crucial role in capturing muscle activity with high precision across various experimental and clinical settings. These devices can be broadly categorized into portable wireless systems, HD-sEMG systems, and integrated wearable solutions. Portable wireless systems offer flexibility for dynamic movements, HD-sEMG provides a detailed spatial mapping of muscle activity, and wearable solutions combine electromyographic data with motion tracking for comprehensive monitoring. The reviewed studies employ a diverse range of sEMG and HD-sEMG systems, each offering unique features tailored to specific research and application needs. Below is a summary of the most commonly used devices, highlighting their features and the studies in which they were applied.

-**EMG Trigno (DelSys®, Natick, MA, USA) [57]**, used in [34,39,47,49]. These electrodes are particularly suitable for detailed biomechanical studies due to their high signal quality and minimal noise interference.-**FREEEMG1000 (BTS Bioengineering, Garbagnate Milanese, Italy) [58]**. A wireless system used in [26,38] to perform Sit-to-Stand tests and gait speed analysis. Its high portability and multi-channel capabilities make it ideal for dynamic tasks.-**Ultium EMG (Noraxon Inc.), Scottsdale, AZ, USA [59],** featured in [42,44]. This device offers advanced signal processing and high fidelity, used for analyzing handgrip contractions at various intensities.-**Custom-Made Flexible Multichannel Electrode System, (Zhejiang University, Hangzhou, China)** used in [43]. This system enables the acquisition of sEMG signals during EF and extension, providing high-resolution data for ML applications.-**EMG acquisition device (Sessantaquatro, OT Bioelettronica, Torino, Italy) [60]**, employed in [36]. It represents a versatile choice for experimental studies due to a semi-disposable adhesive grid of 64 electrodes.-**Myoware EMG SEN-13723 (Advancer Technologies, Raleigh, NC, USA) [61] and Raspberry Pi for data acquisition,** reported in [62]. This cost-effective solution is ideal for home rehabilitation and real-time biofeedback applications.-**sEMG Synergy (Nicolet Biomedical, Madison, WI, USA) [63],** reported in [45]. The system’s high sensitivity allows a precise detection of low-amplitude muscle signals, making it effective for early diagnosis of neuromuscular disorders.-**HD-sEMG grid (Mobita, TMSi, Oldenzaal, The Netherlands) [64],** used in [41]. The high-density configuration enables spatially detailed muscle activity mapping, beneficial for research requiring fine-grained muscle activation patterns.

Each device was selected according to its eligibility for the specific objectives of the study. Some of these more relevant devices are shown in Figure 4. In Figure 5, a graphical representation of the distribution of articles versus the used device is reported.

The successful acquisition of sEMG signals relies not only on appropriate recording techniques but also on effective signal processing methods. Once the raw sEMG data are collected, they must undergo various processing steps to remove noise, extract meaningful features, and facilitate further analysis. The following section outlines the signal processing techniques applied in the reviewed studies, emphasizing their role in enhancing data quality and enabling accurate interpretation of muscle activity.

## 5. Signal Processing Methods

Effective signal processing is crucial in sEMG to ensure accurate analysis of muscle activity. Preprocessing steps, such as baseline correction, normalization, and artifact removal, are essential to mitigate noise, motion artifacts, and signal drift [65,66].

### 5.1. Filtering and Preprocessing

Filtering techniques, including high-pass, low-pass, band-pass, and notch filters [66,67,68,69] are commonly employed to isolate the frequency range of interest, typically between 10 and 500 Hz, and to minimize phase lag and preserve the signal integrity. This frequency range encompasses the primary components of the EMG signal, which are typically found between 10 and 500 Hz. By filtering out frequencies below 20 Hz, movement artifacts and baseline noise are reduced, while filtering above 450 Hz helps eliminate high-frequency noise, such as electromagnetic interference [70]. This comprehensive filtering strategy ensures clean, normalized sEMG signals, facilitating accurate feature extraction and reliable classification.

Advanced processing methods, such as empirical mode decomposition (EMD), wavelet transform, and independent component analysis (ICA) [66,67,68,69], allow a more detailed exploration of sEMG signals and enhance classification accuracy in various applications, including rehabilitation and human–machine interfaces.

#### Analysis of the Selected Studies

The filtering and preprocessing techniques used in the selected studies are reported below.

In Li et al. [42], the signals were sampled at a rate of 2000 Hz with a gain of 1000. To condition the signals, the researchers applied notch filters to eliminate noise at integral multiples of 50 Hz and used a 3rd-order Butterworth band-pass filter with a frequency range of 10–500 Hz. Subsequently, a visual inspection was performed to select three-second segments of sub-maximal contractions that exhibited stationarity. For further analysis, the data were segmented into 200 ms windows with a step of 50 ms, ensuring a comprehensive representation of the signal.

In Ma et al. [27], the raw signals were sampled at 1000 Hz and processed using a 4th-order Butterworth band-pass filter with a frequency from 20 to 450 Hz to remove noise and artefacts and retain the frequency range of muscle activity. Such a filter setup ensures that the frequency range of muscle activity of interest is retained while effectively removing low- and high-frequency noise. The data were segmented into 2000 ms windows with a 50 ms overlap, ensuring a detailed temporal analysis of muscle activity.

In Leone et al. [38], the main steps of the preprocessing phase were as follows: (a) noise reduction, (b) EMG enveloping, and (c) data normalization. Reducing baseline noise and signal distortions caused by EMG electrode movements is the aim of the first step, using a 4th-order Butterworth band-pass filter with a frequency range of 20 Hz to 450 Hz to filter the raw data. Subsequently, to make the signals comparable and suitable for further processing, the linear signal envelope was computed through full rectification and low-pass Butterworth filtering (with a cut-off frequency of 10Hz). Finally, MVC normalization was applied to ensure the comparability of signals across subjects. An oversampling strategy Synthesizing Minority Over-sampling Technology (SMOTE) in combination with Edited Nearest Neighbours (ENN) was used to balance the dataset, addressing the issue of class imbalance in the sarcopenia confidence levels.

In Hung et al. [25], sEMG signals were filtered using a band-pass filter (20–450 Hz) to eliminate noise and artifacts, while six-axis G-sensor data capturing accelerations in three directions were processed for movement analysis. Data segmentation involved creating 10 s windows sampled at 30 Hz, generating 10,048 samples in the dataset for further analysis. This preprocessing ensured clean and reliable data for model training.

In Kumar et al. [39], the raw signals were filtered using a Butterworth band-pass filter (20–400 Hz) to remove noise and artifacts. Segmentation was performed based on synchronized inertial data to divide the signals into individual steps or squat cycles. Signals were normalized to the MVC values and rectified to ensure consistency and comparability across subjects.

In Jin et al. [43], the system acquired the signals using an eight-channel flexible hydrogel electrode array. sEMG data were collected at a frequency of 2000 Hz, with the reference voltage set to 2.4 V and the gain set to 6. Raw sEMG data were processed with a band-pass filter (6–400 Hz), then MVC normalization was applied.

In Godoy et al. [26], the signals were acquired with a sampling rate of 1000 Hz. Preprocessing involved applying a zero-phase Butterworth band-pass filter (10–500 Hz) to remove noise and motion artifacts. Assisted segmentation was then applied to isolate muscle activation phases.

In Sung et al. [45], the signals were recorded using surface electrodes placed according to SENIAM guidelines, with a sampling rate of 48 kHz and a band-pass filter applied (100–500 Hz). Signal rectification was performed for each contraction phase.

In He et al. [44], the signals were filtered with a 3rd-order band-pass filter (20–500 Hz) to remove noise, and a notch filter was applied at 50 Hz. Stable amplitude segments of the sEMG data were manually identified, and further segmentation was performed using 200 ms windows with a 50 ms step size. Signal normalization was achieved dividing the extracted features by their respective MVC values to ensure comparability across participants.

In Zhang et al. [40], signals were sampled at 2000 Hz and filtered with a Butterworth band-pass filter (10–500 Hz).

In Imrani et al. [41], the signals were filtered with a band-pass filter (4–400 Hz) and segmented using the Hilbert Transform to identify muscle contraction dynamics. Noise reduction was implemented, and data were averaged across three trials for consistency. Signal normalization ensured comparability between participants.

Once the signals have been adequately processed and denoised, the next crucial step involves extracting meaningful features that can provide insights into muscle function. The various feature extraction techniques used in sEMG analysis are discussed in more detail in the following section.

### 5.2. Feature Extraction and Selection

Feature extraction [66,67,68,69,71] is a critical step in transforming raw sEMG signals into meaningful parameters for analysis. To assess muscle activity, strength, and neuromuscular function, the extracted features are typically classified into time-domain and frequency-domain features. In particular, time-domain features are most commonly used for EMG pattern recognition being easy and quick to calculate, requiring no transformation. Whereas frequency-domain features are normally based on the estimated power spectral density (PSD) of the signal. They also require more computational resources and time compared to those in the time domain.

#### Analysis of the Selected Articles

In Li et al. [42], the authors extracted features from signals using both time-domain and time-frequency domain approaches. The time-domain features included the Root Mean Square (RMS), Mean Absolute Value (MAV), Integrated EMG (iEMG), Waveform Length (WL), Zero Crossing (ZC), and Slope Sign Change (SSC). In addition, time-frequency domain features were derived from Continuous Wavelet Transform (CWT), specifically, Absolute Power (CWT_power), Kurtosis (CWT_kurtosis), and Wavelet Entropy (WE). To ensure comparability across subjects and contraction levels, the extracted features were normalized using MVC normalization on a channel-specific basis. The statistical significance of the feature differences was assessed using non-parametric tests, including the Mann–Whitney U Test and the Wilcoxon Matched Pairs Signed Rank Test. In the sarcopenic group and the healthy group, statistically significant differences were found in all sEMG features between 20% MVC and 50% MVC, with some exceptions, namely, WE and SSC.

In Ma et al. [27], feature extraction focused on both time-domain and frequency-domain metrics. Time-domain features included the Integrated Absolute Value (IAV) and RMS, which reflect muscle contraction intensity and energy, respectively. Frequency-domain features such as the Mean Power Frequency (MPF) and Median Frequency (MF) were calculated from the power spectrum to analyze muscle fatigue and overall activity. An 80-dimensional feature vector was constructed, combining features from multiple channels for classification purposes.

In Leone et al. [38], a wide range of time-domain and time-frequency domain features were extracted from the sEMG signals of four channels. Key features included iEMG, RMS, MAV, and the Averaged Instantaneous Frequency (AIF). To reduce computational complexity and improve classification performance, the Modified Binary Tree Growth Algorithm was used to select the most relevant features. The final feature set consisted of 12 features, including iEMG, RMS, and AIF for all channels.

In Hung et al. [25], feature extraction included key metrics such as RMS for assessing muscle strength, MNF for muscle fatigue, and specific directional acceleration data from the six-axis G-sensor. These features were combined to provide a comprehensive representation of muscle activity and movement patterns, critical for sarcopenia classification. The RMS, MNF, and Y-axis acceleration were found to be the most important features.

In Kumar et al. [39], EMD was applied to the sEMG signals to decompose them into intrinsic mode functions (IMFs), capturing key oscillatory patterns. Time-domain, frequency-domain, and time-frequency domain features were extracted from the raw sEMG and the first five IMFs, resulting in 868 features, including metrics such as the RMS, MDF, and Sample Entropy (SE). The Minimum Redundancy Maximum Relevance (mRMR) technique was used to select the most informative subset of features for classification.

In Jin et al. [43], two key features were extracted: determinism (DET%), derived from Recurrence Quantification Analysis (RQA), and MF. These features reflect, respectively, the synchronization of motor units and frequency compression due to reduced conduction velocity in muscle fibers. Statistical analysis showed that both features were significantly associated with sarcopenia, with DET% identified as a protective factor and MF as a risk factor.

In Godoy et al. [26], eleven sEMG features were extracted, encompassing the time domain (RMS), frequency domain (MNF, MDF, Bandwidth (BW), Spectral Moment Ratio (SMR)), complexity measures (Sample Entropy (SampEn), Spectral Entropy (SpecEn), Higuchi fractal dimension (HFD)), and shape characteristics (probability density function-based measures like the Center Shape Distance (CSD), Left Shape Distance (LSD), and the Right Shape Distance (RSD)). Notably, features from the BF muscle exhibited more significant differences between sarcopenic and control participants compared to the RF muscle. Principal Component Analysis (PCA) was conducted to reduce dimensionality, identifying the following features: SampEn, SMR, HFD, MNF, BW, MDF, SampEm, RMS, and RSD.

In Sung et al. [45], the mean and maximum values of the RMS during EF, EE, KE, and KF were acquired ((MeanRMS(EF), MaxRMS(EF), MeanRMS(EE), MaxRMS(EE), MeanRMS(KE), MaxRMS(KE), MeanRMS(KF), and MaxRMS(KF)). Furthermore, the ratio of the MeanRMS(EF) divided by the mean value of RMS during EF, EE, KE, and KF was calculated, (RatioRMS(EF), RatioRMS(EE), RatioRMS(KE), RatioRMS(KF)). All these features showed strong correlation with sarcopenia parameters.

In He et al. [44], both time-domain and frequency-domain features were extracted from the sEMG signals. The time-domain features included RMS, MAV, iEMG, SSC, ZC, and WL. Frequency-domain features such as MPF and MDF were also calculated. These features were used to generate normalized spider plots representing the muscle coordination patterns across six channels.

In Zhang et al. [40], iEMG was calculated to determine muscle activity intensity, while the amplitude contribution ratio (ACR) and co-contraction ratio (CCR) were derived to assess muscle coordination. Frequency-domain features such as MPF and MF were computed to evaluate muscle fatigue and contraction dynamics.

In Imarani et al. [41], two HD-sEMG indexes were extracted: Muscular Contraction Intensity (MCI) and Muscle Contraction Dynamics (MCD). MCI quantified the amplitude of muscle activation, while MCD reflected the duration of contractions. These features were computed for each electrode channel and averaged across trials to provide robust measures of muscle function. Both indexes were statistically analyzed to assess their relationship with age and physical activity levels.

## 6. Statistical Analysis and Artificial Intelligence Approaches for Sarcopenia

Statistical Analysis (SA) and Artificial Intelligence (AI) are powerful tools for advancing sarcopenia detection, diagnosis, and assessment or sarcopenia-related parameters by analyzing the extracted features from sEMG signals. In fact, through the automatic extraction and interpretation of meaningful patterns, AI facilitates the early identification of muscle mass loss, functional impairments, and low muscle quality associated with sarcopenia. This section explores the key AI approaches and the SA methods used in this field.

### 6.1. Analysis in the Selected Studies

In Li et al. [42], the authors employed ML techniques to classify sarcopenic and healthy individuals based on the extracted sEMG features. A voting classifier was developed by combining Support Vector Machine (SVM), Random Forest (RF), and Gradient Boosting Machine (GBM) models. The voting classifier achieved an average accuracy of 73% and an average sensitivity of 79% through five-fold cross-validation, outperforming the individual models.

In Ma et al. [27], six ML classifiers were implemented and compared: SVM, Decision Tree (DT), RF, Logistic Regression (LR), K-Nearest Neighbors (KNN), and Naive Bayes (NB). The classifiers were trained on labeled data to distinguish between sarcopenic and healthy subjects. Among these, SVM emerged as the most effective, leveraging its ability to identify optimal hyperplanes for classification in high-dimensional feature spaces, obtaining the highest classification accuracy (87.5%), outperforming DT (75%), RF (62.5%), KNN (62.5%), NB (75%), and LR (50%). The study concluded that ML, particularly SVM, can serve as a reliable tool for non-invasive, cost-effective sarcopenia screening, with potential for broader applications in clinical and homecare settings.

In Leone et al. [38], eight supervised ML classifiers were compared to evaluate their ability to classify three sarcopenia confidence levels on 32 patients recruited from a hospital. These classifiers included SVM, DT, RF, LR, KNN, NB, Multi-layer Perceptron (MLP), and Extreme Gradient Boosting (XGB). The classifiers were trained and tested using a ten-fold cross-validation approach. Feature selection significantly improved model performance, with SVM showing the highest accuracy and robustness, achieving an accuracy of 96.7%, precision of 96.9%, recall of 95.3%, and an F1-score of 95.6% after feature selection, resulting in an improvement of approximately 10% in accuracy versus all features. Other classifiers, such as KNN and XGB, also performed well, achieving over 90% accuracy. The study demonstrated the effectiveness of the proposed pipeline in distinguishing sarcopenia confidence levels and highlighted the importance of feature selection and dataset balancing in improving classifier performance.

In Hung et al. [25], five ML algorithms (RF, DT, SVM, KNN, NB) were trained using a 70/30 train–test split. A confusion matrix, precision, recall, and the F1-score were employed to evaluate model performance. RF emerged as the most effective algorithm, achieving the highest accuracy and generalization capabilities. The RF classifier achieved the best accuracy of 96.37%, followed by DT (93.13%), KNN (89.65%), SVM (81.56%), and NB (75.52%). The results demonstrated the system’s effectiveness in classifying sarcopenia severity. The ROC curve is used to evaluate classifier performance, and the area under the ROC curve (AUC) is an essential metric for measuring classifier performance, where higher AUC values indicate better classifier performance. High AUC values across all classifiers further supported their reliability. The study concluded that wearable sensor-based systems combined with ML provide a promising approach for non-invasive sarcopenia diagnosis and monitoring in elderly populations.

In Kumar et al. [39], five ML algorithms were employed to classify individuals into healthy and sarcopenia-risk groups: KNN, NB, RF, XGB, and MLP. To evaluate the performance of the classifier, Leave-One-Subject-Out (LOSO-CV) cross-validation was employed and the hyperparameters were optimized with GridSearchCV. The MLP model consistently outperformed others due to its ability to learn complex, non-linear relationships. The MLP classifier demonstrated the highest accuracy in detecting sarcopenia risk across all physical activities, achieving 88% for normal walking, 89% for fast walking, 81% for standard squats, and 80% for wide squats. The integration of EMD-based features improved model accuracy by 4–5% compared to models without EMD decomposition. These results highlight the effectiveness of the proposed sEMG-EMD-ML pipeline for early sarcopenia risk detection, providing a robust tool for analyzing muscle function and potential dysfunction during daily activities.

In Jin et al. [43], a system based on Binomial Logistic Regression analysis was used to evaluate the effectiveness of the extracted features in diagnosing sarcopenia. Data were collected through the sEMG sensor system and analyzed using the Statistical Package for the Social Sciences. Tested on 81 elderly subjects, the system identified two significant features: DET% as a protective factor (OR = 0.711) and MF as a risk factor (OR = 1.032). The system demonstrated potential for cost-effective, home-based sarcopenia screening.

In Godoy et al. [26], linear discriminant analysis (LDA) was performed on the most significant features (SampEn and MDF) from the BF muscle to classify sarcopenic and control participants. Cross-validation using the Leave-One-Out method yielded an 88.9% classification accuracy, demonstrating the robustness of the selected biomarkers. While the study did not implement advanced ML algorithms, the stepwise refinement of features ensured optimal discrimination. Significant differences were observed in BF muscle sEMG parameters between sarcopenic and control groups, particularly in spectral (MDF, MNF) and complexity (SampEn, SpecEn) features. The sarcopenic group exhibited lower MNF, MDF, and complexity values, indicative of reduced motor unit recruitment and altered muscle function. The LDA model successfully classified participants with high accuracy, highlighting the potential of SampEn and MDF as biomarkers for sarcopenia diagnosis. These findings emphasize the utility of sEMG in developing non-invasive, reliable tools for early sarcopenia assessment.

In Piasecki et al. [37] the aim of the study was to compare motor unit size and number between young (*n* = 48), non-sarcopenic old (n=13), pre-sarcopenic (*n* = 53), and sarcopenic (*n* = 29) men. Motor unit potentials (MUPs) were isolated from intramuscular and sEMG recordings. By statistical analyses (ANOVA), the authors showed that the motor unit numbers were reduced in all groups of old compared with young men (all *p* < 0.001). MUPs were elevated in non-sarcopenic and pre-sarcopenic males relative to young men (*p* = 0.039 and 0.001, respectively), but not in the VL of sarcopenic older individuals (*p* = 0.485). The results suggest that extensive motor unit remodeling occurs relatively early during ageing, exceeds the loss of muscle mass, and precedes sarcopenia. The reinnervation of denervated muscle fibers likely increases motor unit size in non-sarcopenic and pre-sarcopenic elderly individuals, but not in those with sarcopenia, indicating that the inability to enlarge motor unit size differentiates sarcopenic from pre-sarcopenic muscles.

In Hirono et al. [36], statistical analyses, including ANCOVA and Kruskal–Wallis tests, were used to compare motor unit firing patterns between normal and pre-sarcopenic older adults. In the normal group, firing patterns exhibited a hierarchical structure consistent with the “onion skin phenomenon” where lower-threshold motor units fired at higher rates. In contrast, the pre-sarcopenic group lacked this hierarchy, with irregular firing patterns observed. Muscle mass, thickness, and echo intensity were significantly lower in the pre-sarcopenic group. The most significant results concerning the EMG features were for the medium-Motor Unit and high-Motor Unit recruitment threshold, with *p*-values less than 0.001, indicating highly significant differences between groups. The findings underscore the presence of early neuromuscular alterations in pre-sarcopenic individuals, suggesting the potential of HD-sEMG for non-invasive assessment and early intervention to prevent sarcopenia progression.

In Sung et al. [45], stepwise linear regression was employed to develop a prediction model for Appendicular Skeletal Muscle mass (ASM) based on sEMG parameters and demographic data (height, weight, sex). The final equation incorporated MeanRMS(EE) and RatioRMS(KF) as significant predictors alongside physical measurements. The model demonstrated a high adjusted $R^{2}$ of 0.934, validating its accuracy through cross-validation on an independent dataset. While no advanced AI models were applied, the regression-based approach provided robust predictions for muscle mass estimation. The regression model showed a strong correlation (R = 0.967, *p* < 0.001) between predicted ASM values and BIA-measured ASM, with minimal prediction error. sEMG parameters correlated positively with muscle strength and mass, but negatively with fat mass, supporting their relevance for muscle assessment. The study concluded that sEMG-derived metrics are reliable indicators of muscle health and can serve as practical tools for sarcopenia diagnosis and monitoring in clinical and homecare settings.

In He et al. [44], a novel index, the Incenter-Circumcenter Distance of Muscle Coordination (ICDMC), derived from the symmetry analysis of spider plots was introduced, quantifing the coordination and distribution of muscle activity patterns. While no ML models were explicitly applied, the ICDMC served as a sophisticated metric to differentiate sarcopenic from healthy individuals based on sEMG features. In addition, significant differences were observed in ICDMC values between sarcopenic and healthy groups, particularly at lower contraction levels (20% MVC). Time-domain features, such as RMS, MAV, SSC, and WL, showed strong discriminative power, with the sarcopenic group displaying more consistent and symmetric muscle coordination patterns. Gender-specific differences were noted, with distinct feature significance observed in males and females. The obtained results support the potential of sEMG and the ICDMC metric as effective tools for sarcopenia screening in community settings.

In Zhang et al. [40], statistical analyses including two-way ANOVA and LR were employed to examine group differences and predictive associations between sEMG-derived metrics and sarcopenia. Results revealed that older adults with sarcopenia exhibited higher muscle activity intensity (iEMG) but reduced coordination (lower CCR) compared to controls. Sarcopenia was also associated with lower contraction frequencies (MPF and MF) in the nondominant GM, indicating greater fatigue vulnerability. These metrics showed moderate diagnostic power (AUC = 0.604–0.663, *p* < 0.01). The findings suggest that sEMG-based assessments can effectively capture sarcopenia-related postural and neuromuscular impairments, providing insights for targeted interventions to improve balance and reduce fall risk.

In Imrani et al. [41], statistical hypothesis testing and multivariate analysis of variance (MANOVA) were used to evaluate the sensitivity of HD-sEMG indexes to age- and activity-related muscle changes. No ML algorithms were employed; however, the HD-sEMG system itself functioned as a diagnostic tool, providing clinicians with intuitive and reliable metrics for muscle health assessment. The integration of these metrics into a broader clinical framework was emphasized. HD-sEMG indexes successfully differentiated between age groups and physical activity levels. MCI and MCD values correlated strongly with aging (R = 0.58 and R = 0.81, respectively). Notably, sedentary individuals with an age range of 45–54 years exhibited HD-sEMG profiles similar to those of older active participants (55–75 years), suggesting that inactivity accelerates muscle ageing. HD-sEMG outperformed traditional clinical parameters (e.g., handgrip strength and DXA measures) in detecting early signs of muscle decline, highlighting its potential as a non-invasive tool for sarcopenia prevention and monitoring.

In Kienbacher et al. [47], two-factorial ANOVAs served to examine the age and gender-specific effects, and models from Generalizability Theory (G-Theory) were used for assessing retest-reliability. The maximum back extension moment was non-significantly smaller in elders. Initial MF (IMF) was overall higher in elders, while MF fatigue declines were significantly smaller in L5 in the recording with the most negative slope, or if the slope of all electrodes was considered. The reliability of the retest was consistent among both young and old individuals. The ICC type measurements of IMF Theory G slopes and fatigue ranged from 0.7 to 0.85. For the fatigue slope declines, absolute SEM values were found to be quite high, but clinically acceptable for the Initial MF. The MF fatigue method can elucidate alterations of ageing back muscles. This method, thus, might be suggested as a potential biomarker to objectively identify persons at risk of sarcopenia. Spectral EMG may potentially be utilized as an outcome-monitoring technique in older populations due to the clinical significance of the IMF in comparison to the MF slope reductions.

The graphical representation of the article distribution versus the ‘classification’ method is shown in Figure 6.

### 6.2. Interpretability and Clinical Feasibility of AI Models

Sarcopenia is not easy to diagnose, but the scientific literature has highlighted the usefulness of Decision Support Systems (DSSs) that facilitate the task of caregivers or clinicians to diagnose the condition or monitor progress or time declines. In particular, the main objective of the analyzed studies, dealing with the detection of sarcopenia using AI methods, was to design and deploy platforms integrating sEMG technology and software modules similar to DSSs for healthcare personnel. In this way, the various systems can provide useful additional knowledge in a cost-effective and non-invasive approach to evaluate the user’s muscle condition during performance tests. Furthermore, being non-invasive and easy-to-use systems, they can also easily operate in a nursing home or home environment to support more frequent follow-up than hospital-based investigations. This allows sarcopenia or muscle decline risk indices to be provided in the prevention phase to help medical staff in evaluating an individual’s risk level and monitoring disease progression. Furthermore, it is helpful to consider the different performances achieved in the analyzed studies. In particular, the classifiers that performed best are SVM with 96.7% in Leone et al. [38] and RF with 96.37% in Hung et al. [25]. However, a very important observation is that these performances were obtained with different datasets, so it is not possible to assert absolutely that, for example, SVM is superior to RF, as the result obviously depends on the available dataset. To obtain these results, it is important to note that several features extracted from the sEMG signal were used to train the used classifiers. Specifically, in Leone et al. [38], some features were extracted in the time domain and time-frequency domain: iMEG, MAV, RMS, and AIF. From these features, three features (iEMG, RMS, AIF) were selected for the four channels by feature selection algorithms (Modified Binary Tree Growth Algorithm) so as to reduce the computational complexity of the employed algorithms. In contrast, in Hung et al. [25], the extracted features in the time domain and frequency domain were the RMS, MNF, and MDF, and those selected were the RMS and MNF. In general, the literature review shows that the most frequently used features for the detection of sarcopenia appear to be the RMS, iEMG, MAV, SSC, MPF, and MF.

## 7. Other Studies

In recent years, numerous studies have focused on identifying the presence of sarcopenia, assessing the risk of its development, and determining features closely associated with the condition. This section provides an overview of relevant research in line with these objectives, offering insights into various approaches and methodologies. In particular, studies using electromyographic signals to analyze muscle fatigue, grip strength, and overall muscle strength, as well as other related parameters, are discussed. Additionally, studies that assess various physiological and functional muscle parameters in both older and younger individuals through sEMG to predict muscle decline over time are reported. These studies contribute to a better understanding of the potential biomarkers and assessment techniques critical for the early detection and management of sarcopenia.

In Song et al. [46], a wearable device leveraging sEMG and electrical stimulation was developed to assess muscle function and facilitate at-home sarcopenia monitoring. The system, tested on 98 healthy participants, showed a strong correlation (r = 0.89) between estimated and actual muscle function, demonstrating its feasibility for non-invasive muscle assessment.

Similarly, Fialkoff et al. [62] introduced a framework to estimate handgrip force using sEMG-based inverse EMG imaging, achieving high accuracy (r = 0.95, RMSE = 0.18N). Despite promising results, further validation is required for clinical application.

In Watanabe et al. [72], multi-channel sEMG was used to examine neuromuscular activation patterns in elderly individuals, revealing that muscle strength is influenced not only by muscle volume but also by neuromuscular efficiency. These findings support sEMG as a tool for sarcopenia diagnosis and rehabilitation planning.

Wu et al. [73] assessed neuromuscular and mechanical declines in younger and older adults, highlighting sex-related differences in age-related muscle strength loss. In [74], reductions in motoneuron excitability of the soleus and TA muscles were found in older adults, providing insights into neuromuscular impairments and potential intervention targets.

Further research, such as Ebenbichler et al. [49] and Wu et al. [75], investigated age-related fatigability and neuromuscular changes, demonstrating that younger individuals exhibit higher glycolytic muscle capacity and greater resistance to fatigue compared to older participants. In Ou et al. [76], sEMG-based probability density function features were used to detect exercise-induced muscle fatigue in community settings, offering a practical approach for real-time monitoring.

Finally, Huang et al. [48] evaluated the effects of a 12-week Tai Chi intervention on neuromuscular responses and postural control in elderly sarcopenic patients. The intervention significantly improved muscle response times and dynamic postural control, reducing fall risk.

These studies collectively emphasize the growing role of sEMG in assessing muscle health, identifying biomarkers, and developing interventions to manage sarcopenia in ageing populations.

## 8. Discussion

Sarcopenia, characterized by progressive muscle mass loss, functional decline, and increased frailty, remains a significant health challenge. sEMG emerges as a valuable tool for sarcopenia diagnosis/detection or assessment due to its capability to capture neuromuscular information. This review analyzed 18 studies investigating the use of sEMG for sarcopenia detection, diagnosis, and monitoring. However, while sEMG offers advantages over traditional diagnostic methods, its clinical applicability remains to be fully validated.

The findings show that lower limb muscles, such as the VL, GL, TA, RF, and BF, are often targeted for sEMG signal acquisition due to their critical role in mobility and balance, which are central in the assessment of sarcopenia. Additionally, upper limb muscles, including the BRA and BB, have been explored recently for their relevance to grip strength, a key diagnostic marker.

Regarding motor tasks, both isometric and dynamic exercises are widely used. Isometric tasks, such as MVC tests and handgrip exercises, provide insights into muscle activation levels and strength. Dynamic tasks, including Sit-to-Stand tests, gait analysis, and walking assessments, are employed to evaluate functional mobility and endurance in daily activities. Lower limb muscles and dynamic tasks are more frequently studied due to their direct impact on mobility and fall prevention in ageing populations. However, upper limb and core muscle studies are gaining attention for comprehensive functional assessments. However, the heterogeneity of methodologies and lack of standardized protocols complicate the comparison and interpretation of findings across studies. Remarkably, while these muscle groups and tasks (e.g., Sit-to-Stand tests, gait speed evaluations, isometric contractions) are commonly studied, there is no consensus on standardized protocols.

Previous studies on sEMG in neuromuscular disorders have shown its utility in assessing muscle activation, coordination, and fatigue. While sEMG has been successfully used in clinical assessments of conditions such as Parkinson’s disease and post-stroke rehabilitation, its application in sarcopenia remains underexplored. Few studies directly compare sEMG measurements with standard sarcopenia diagnostic tools, such as DXA or handgrip strength, limiting our ability to draw strong clinical correlations.

An analysis of the sEMG devices used shows that wireless systems, such as the BTSFreeEMG1000, due to its portability are preferred for dynamic tasks, while high-resolution spatial systems, such as HD-sEMG electrodes, are more suitable for advanced research applications.

The most commonly used signal processing techniques in sEMG studies include Butterworth band-pass filtering, which is frequently applied with cutoff frequencies typically ranging between 10–500 Hz to remove low-frequency noise and high-frequency interference. Butterworth filters are favored due to their smooth frequency response and minimal signal distortion, making them ideal for physiological signals. Notch filters, often set at 50 or 60 Hz, are used to eliminate power line interference. Many studies also employ segmentation techniques, such as dividing signals into fixed time windows (e.g., 200 ms with 50 ms overlap), to facilitate detailed analysis of muscle activation patterns. Normalization methods, particularly MVC, are widely used to ensure consistency and comparability across subjects. Additionally, rectification and envelope extraction using low-pass filtering (e.g., 10 Hz) are applied to prepare the signals for further analysis. However, the choice of cutoff frequencies can impact the detection of relevant muscle activity.

The features in sEMG studies are primarily categorized into time-domain and frequency-domain features, with some studies also incorporating time-frequency and complexity-based features to enhance muscle activity assessment. Time-domain features, such as RMS, MAV, iEMG, ZC, and SSC, are widely used due to their simplicity and computational efficiency. They provide valuable insights into muscle contraction intensity, endurance, and overall activity and can be used also for real-time application due to their low computational cost. Commonly extracted frequency-domain metrics include MPF and MF, which offer insights into muscle fatigue and fiber type composition. Studies focusing on fatigue detection and muscle coordination often rely on frequency-domain features for a deeper understanding of signal behavior over time.

Some studies use CWT and EMD to analyze dynamic muscle activation patterns, providing improved resolution in time and frequency, and helping to detect transient changes in muscle activity. WE and complexity measures such as SampEn are increasingly used for feature extraction, particularly in machine learning-based classification tasks. These techniques are less commonly used due to their computational complexity.

The increasing integration of ML techniques in sarcopenia assessment highlights the need to select the most important features. In fact, feature selection techniques, such as mRMR and PCA, have been employed to reduce data dimensionality while retaining essential signal information. Studies show that combining multiple feature types—such as RMS and MF—provides better classification accuracy and robustness in detecting sarcopenia-related changes.

In addition, the reviewed studies employed a range of AI and statistical techniques for sarcopenia detection, classification, and evaluation based on extracted sEMG features. Commonly used models include SVM, RF, KNN, DT, and LR. SVM consistently emerged as a top performer, achieving accuracy rates of up to 96.7%, particularly when feature selection techniques were applied. RF has shown robust performance, achieving high accuracy (96.37%) and generalization capabilities across studies. Deep Learning methods, used in some related studies, while promising, are still under development and require larger datasets for robust generalization. They offer the advantage of automatic feature extraction but are computationally time-consuming and overfitting with small datasets.

Traditional statistical techniques, including ANOVA, regression models, and SHAP, have been widely used to analyze sEMG features and identify significant indicators of sarcopenia. Regression models have shown strong predictive capabilities, effectively correlating sEMG metrics with clinical measures such as muscle mass and strength. Statistical analysis methods remain a valuable complement to AI-based approaches, providing interpretable insights into muscle function and aiding in feature selection to improve model performance. The integration of AI with traditional statistical approaches enhances the diagnostic potential of sEMG.

The review highlighted that new lines and approaches of research on the assessment of sarcopenia through the analysis of electromyographic signals have been developed in recent years, and the first interesting results have been published. However, several challenges need to be addressed to enable this technology to be efficient and suitable for the diagnosis and monitoring of disease progression. One significant issue is the variability in electrode placement, muscle activation patterns, and subject-specific factors, introducing potential inconsistencies in sEMG recordings. Additionally, many studies are conducted with relatively small sample sizes, limiting the generalizability of the findings. Although static tasks often produce reliable results, the presence of motion artifacts during dynamic movements complicates signal analysis. Furthermore, differences in preprocessing techniques and feature extraction methods between the various studies present reproducibility and comparability problems, making it difficult to establish standardized protocols. Moreover, although promising, most studies have been performed in controlled laboratory conditions rather than real-world clinical scenarios. Finally, current research mainly focuses on cross-sectional analysis, limiting our understanding of sEMG’s potential for tracking sarcopenia progression over time.

The findings of this systematic review suggest that sEMG could play a role in the future of sarcopenia diagnosis, particularly in the development of non-invasive, quantitative assessment tools. However, several key areas need to integrate sEMG into clinical practice. Establishing guidelines for electrode placement, signal preprocessing, and feature selection would improve reproducibility. Conducting multi-center studies with larger sample sizes and diverse populations is essential to improve the robustness and generalizability of the models. Additionally, the integration of sEMG with devices (such as socks, bands) could enable the development of portable and real-time monitoring systems, facilitating long-term tracking of muscle health in both clinical and home settings. Another important aspect is the adoption of hybrid approaches combining sEMG with other physiological signals, such as accelerometry and force sensors, to provide a more comprehensive assessment of muscle function. Finally, the development of cloud-based or edge AI solutions would allow real-time data processing, offering immediate clinical feedback and improving decision- making processes.

## 9. Conclusions

This review examined the use of sEMG for the diagnosis and assessment of sarcopenia, focusing on the muscles involved, signal processing techniques, extracted features, and classification systems, including AI and statistical analysis. Through a rigorous selection process, relevant scientific publications were analyzed, excluding inconsistent or low-quality studies. Additionally, a comprehensive understanding of the current landscape of commercial and non-commercial sEMG devices for the assessment of muscle function was also provided.

An analysis of publication dates shows a growing interest in this topic, with most papers published in the last decade and a significant increase in the last five years. This trend is driven by the need for non-invasive, cost-effective diagnostic tools to facilitate early detection and monitoring.

To contribute to the academic debate and serve as a valuable resource for researchers, clinicians, and technology developers, key challenges were identified and potential directions for future research and technological innovation were suggested.

## Figures and Tables

**Figure 1 sensors-25-02122-f001:**
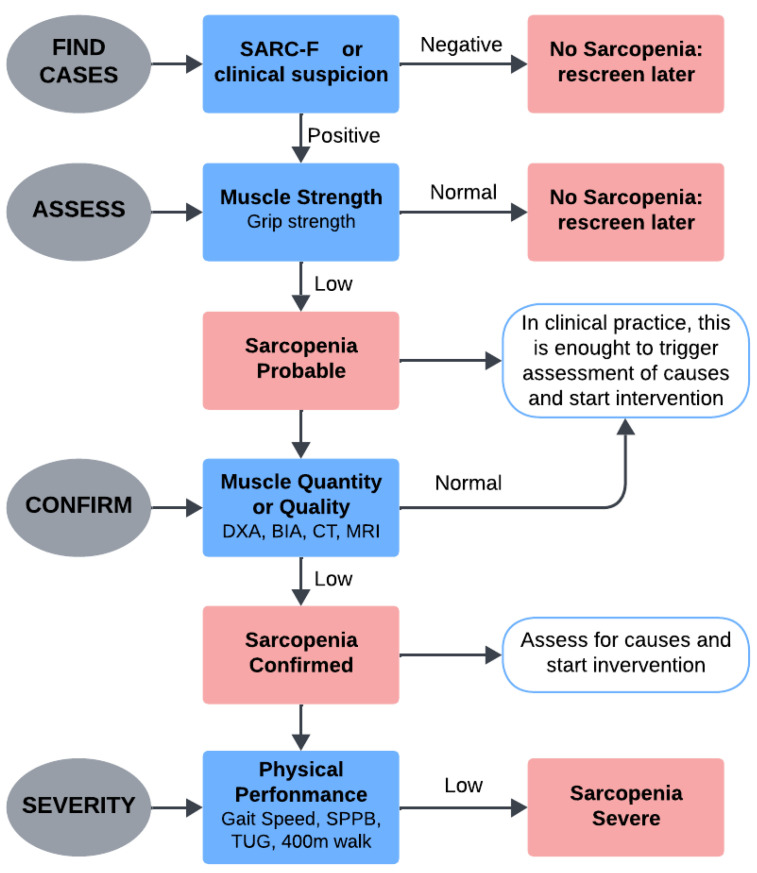
Algorithm to diagnose sarcopenia in clinical practice.

**Figure 2 sensors-25-02122-f002:**
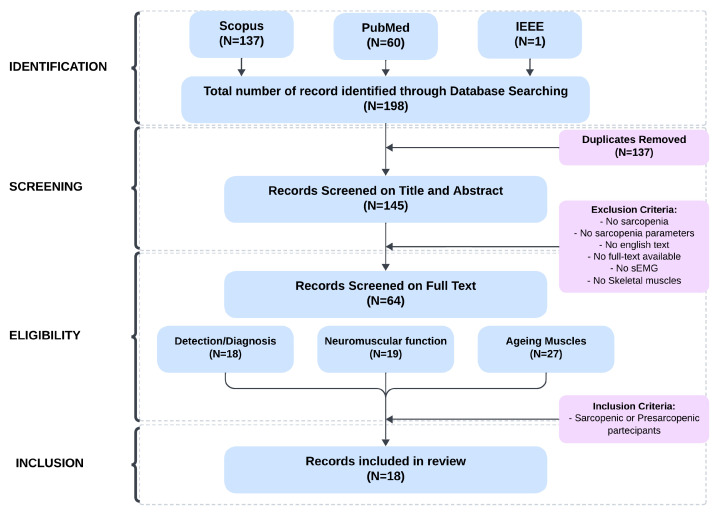
Flow diagram generated with PRISMA-S methodology, depicting the reviewers’ process of finding published data on the considered topic and how they decided whether to include it in the review.

**Figure 3 sensors-25-02122-f003:**
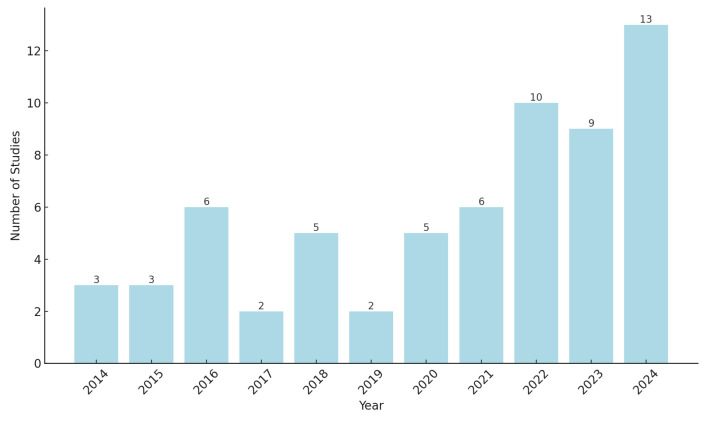
Distribution of the articles by year of publication.

**Figure 4 sensors-25-02122-f004:**
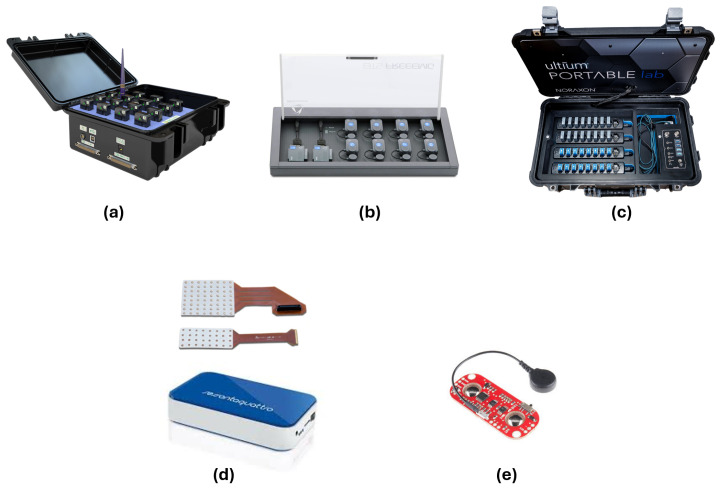
Most used devices: (**a**) Delsys Trigno Wireless System; (**b**) BTSFreeEMG1000; (**c**) Noraxon Ultium EMG; (**d**) HD flexible electrode and Sessantaquattro by OT Bioelectronica device; (**e**) Myoware EMG SEN-13723.

**Figure 5 sensors-25-02122-f005:**
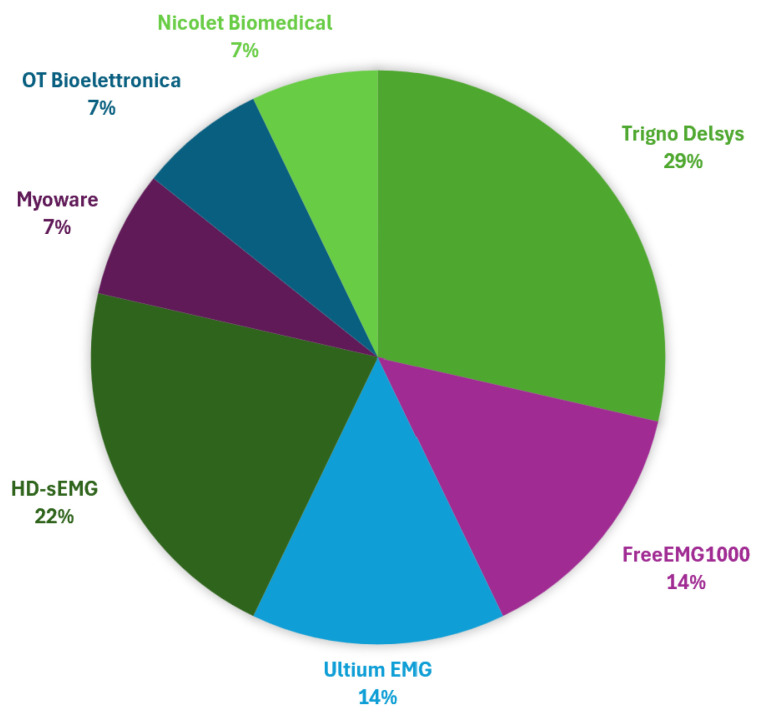
Graphical representation of the distribution of articles with respect to used device.

**Figure 6 sensors-25-02122-f006:**
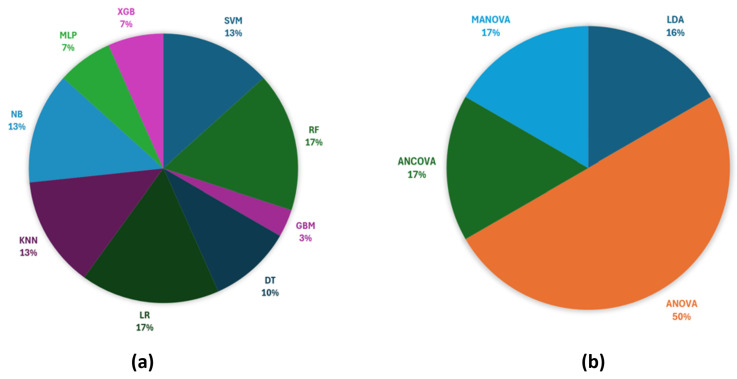
Graphical representation of the artificial intelligence (**a**) and statistical analysis methods (**b**) used to detect/diagnose sarcopenia.

**Table 1 sensors-25-02122-t001:** Search strategy for each considered multidisciplinary database.

Database	Query
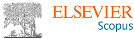	TITLE-ABS-KEY (“surface electromyography” OR “sEMG” OR “surface EMG” OR “Superficial EMG” OR “Superficial Electromyography” OR “s-EMG” OR “Surface-based EMG”) AND TITLE-ABS-KEY (“sarcopenia” OR “presarcopenia” OR “muscle strength” OR “muscle mass” OR “muscle quality” OR “neuromuscular activation” OR “Age-related muscle loss” OR “Muscle wasting” OR “Age-associated sarcopenia” OR “Muscle atrophy” OR “muscle mass loss” OR “Muscle strength testing” OR “Handgrip dynamometry” OR “Grip strength”) AND TITLE-ABS-KEY (“elderly” OR “aging population” OR “older people” OR “older adults” OR “older individuals” OR “elderly population”)
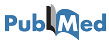	((surface electromyography[Title/Abstract]) OR (sEMG[Title/Abstract]) OR (surface EMG[Title/Abstract]) OR (Superficial EMG[Title/Abstract]) OR (Superficial Electromyography[Title/Abstract]) OR (s-EMG[Title/Abstract]) OR (Surface-based EMG[Title/Abstract])) AND ((sarcopenia[Title/Abstract]) OR (presarcopenia[Title/Abstract]) OR (muscle strength[Title/Abstract]) OR (muscle mass[Title/Abstract]) OR (muscle quality[Title/Abstract]) OR (neuromuscular activation[Title/Abstract]) OR (Age-related muscle loss[Title/Abstract]) OR (Muscle wasting[Title/Abstract]) OR (Age-associated sarcopenia[Title/Abstract]) OR (Muscle atrophy[Title/Abstract]) OR (muscle mass loss[Title/Abstract]) OR (Muscle strength testing[Title/Abstract]) OR (Handgrip dynamometry[Title/Abstract]) OR (Grip strength[Title/Abstract])) AND ((elderly[Title/Abstract]) OR (aging population[Title/Abstract]) OR (older people[Title/Abstract]) OR (older adults[Title/Abstract]) OR (older individuals[Title/Abstract]) OR (elderly population[Title/Abstract]))
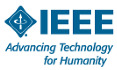	(((“Document Title”:sEMG) OR (“Document Title”:“surface electromiography”) OR (“Document Title”:“surface EMG”) OR (“Document Title”:“Superficial EMG”) OR (“Document Title”:“Superficial Electromyography”) OR (“Document Title”:“Surface-based EMG”) OR (“Document Title”:“s-EMG”)) AND ((“Document Title”:“sarcopenia”) OR (“Document Title”:“presarcopenia”) OR (“Document Title”:“muscle strenght”) OR (“Document Title”:“muscle mass”) OR (“Document Title”:“muscle quality”) OR (“Document Title”:“neuromuscular activation”) OR (“Document Title”:“age-related muscle loss”) OR (“Document Title”:“Muscle wasting”) OR (“Document Title”:“Age-associated sarcopenia”) OR (“Document Title”:“Muscle atrophy”) OR (“Document Title”: “muscle mass loss”) OR (“Document Title”:“Muscle strength testing”) OR (“Document Title”:“Handgrip dynamometry”) OR (“Document Title”:“Grip strength”)) AND ((“Document Title”:“elderly”) OR (“Document Title”:“aging population”) OR (“Document Title”:“older people”) OR (“Document Title”:“older adults”) OR (“Document Title”:“older individuals”) OR (“Document Title”:“elderly population”)) AND ((“Abstract”:sEMG) OR (“Abstract”:“surface electromiography”) OR (“Abstract”:“surface EMG”) OR (“Abstract”:“Superficial EMG”)) AND ((“Abstract”:“sarcopenia”) OR (“Abstract”:“presarcopenia”) OR (“Abstract”:“muscle strenght”) OR (“Abstract”:“muscle mass”) OR (“Abstract”:“muscle quality”) OR (“Abstract”:“neuromuscular activation”) OR (“Abstract”:“age-related muscle loss”) OR (“Abstract”:“Muscle wasting”) OR (“Abstract”:“Age-associated sarcopenia”) OR (“Abstract”:“Muscle atrophy”) OR (“Abstract”: “muscle mass loss”) OR (“Abstract”:“Muscle strength testing”) OR (“Abstract”: “Handgrip dynamometry”) OR (“Abstract”:“Grip strength”)) AND ((“Abstract”:“elderly”) OR (“Abstract”:“aging population”) OR (“Abstract”:“older people”) OR (“Abstract”:“older adults”) OR (“Abstract”:“older individuals”) OR (“Abstract”:“elderly population”)))

**Table 2 sensors-25-02122-t002:** Muscles and motor tasks.

Ref.	Muscles	Motor Tasks
Hirono et al. [36]	VL	MVC of knee extensors
Pasecki et al. [37]	VL, TA	MVC of knee extensors
Leone et al. [38]	GL, TA	Sit-to-Stand test, gait speed test (5 m)
Hung et al. [25]	Not explicitly specified, but focused on lower limb muscles	Rehabilitation exercises prescribed by physiotherapists for lower limbs
Kumar et al. [39]	RF, BF, TA, and GL	Normal walking, fast walking, standard squat, wide squat
Zhang et al. [40]	GM, RF, BF, TA, GL	Static standing posture
Imrani et al. [41]	RF	3 times Sit-To-Stand (chair rising)
Godoy et al. [26]	RF, BF	Sit-To-Stand test for 30 s without using hands
Li et al. [42]	BRA, FCR, FDS, FCU, E CU, ED	Contractions at 20% and 50% of MVC during handgrip tasks
Jin et al. [43]	BB	MVC and SVC during EF, EE
He et al. [44]	BRA, FCR, FDS, FCU, ECU, ED	Handgrip contractions at 20% and 50% of MVC
Sung et al. [45]	BB, TB, RF, and BF	MVC during EF, EE, KF, KE
Ma et al. [27]	L5, L1	Lumbar movements (forward bends, lateral bends), maximum handgrip test, Sit-to-Stand test

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
