# Peer review of "A Systematic Review of Surface Electromyography in Sarcopenia: Muscles Involved, Signal Processing Techniques, Significant Features, and Artificial Intelligence Approaches"

_sensors, 2025, doi:10.3390/s25072122_

Round 1
Reviewer 1 Report
Comments and Suggestions for Authors
Comments:
This study reviewed recent literature and attempted to evaluate the application of surface EMG in sarcopenia detection and diagnosis. The Methods section of the manuscript was clearly written, and it provided detailed descriptions of the articles it included. However, I think the rationale/need for the study needs to be clearly addressed. Also, the summary of the review, its conclusion, and application are not clear enough for the readers. Below are my comments and questions:
- The Rationale and need for the study was not clearly addressed. The authors argued that traditional methods, such as DXA, MRI, have limited accessibility and require specialized equipment and personnel to operate, particularly in resource-limited settings or for early diagnosis. EMG also requires specialized equipment and personnel to operate. As for early detection of sarcopenia, different methods could detect different “signs” in the muscles, I don’t think it is fair to say that EMG can perform early detection but DXA or MRI cannot.
- It seems conference papers were included in this review. I suggested the authors clearly indicate which articles are full papers and which are conference papers. Conference papers are usually preliminary studies and have not gone through rigorous peer review, so their scientific evidence level is not at the same level as a research paper.
- General suggestion: when citing a paper in the text, address the article with the last name of the first author, rather than simply using “In [34], sEMG signals…”. This also applies to the tables.
- I do not think it is important information to go into detail and list all the EMG equipment, their figures, and the distribution of articles that used that equipment (Figures 4 & 5).
- Tables 3, 4 & 5 provided the same information as the corresponding paragraph. I suggest the authors rethink what information to put in the text and the tables and avoid redundancy. Also, the paragraphs in the Results section mainly describe each individual paper, but I think if the authors can provide a more comprehensive summary and analysis of all the papers, it will be more beneficial to the readers.
- It is unclear why statistical analysis and artificial intelligence were put together in Figure 6 for comparison.
- The Discussion section mainly describes the findings again, but the summary/major findings of this review paper were clearly stated. From this review, there are many studies attempted to use EMG for sarcopenia detection and diagnosis, a clear statement of what has been found and EMG can or cannot do for this application needs to be thoroughly and clearly add
Author Response
We thank the reviewer for the received feedbacks.
- Question: The Rationale and need for the study was not clearly addressed. The authors argued that traditional methods, such as DXA, MRI, have limited accessibility and require specialized equipment and personnel to operate, particularly in resource-limited settings or for early diagnosis. EMG also requires specialized equipment and personnel to operate. As for early detection of sarcopenia, different methods could detect different “signs” in the muscles, I don’t think it is fair to say that EMG can perform early detection, but DXA or MRI cannot.
Response: We thank the reviewer for the comment. sEMG is a technology that, although it requires specialized personnel, does not necessarily require trips to hospital facilities as some commercial solutions can be used directly at home. We have modified the introduction section (line 92-95, line 98-102) by making the comparison between sEMG and traditional methods fairer by explicating the limitations of sEMG, then we made the purpose of the review clearer (line 110-126)
- Question: It seems conference papers were included in this review. I suggested the authors clearly indicate which articles are full papers and which are conference papers. Conference papers are usually preliminary studies and have not gone through rigorous peer review, so their scientific evidence level is not at the same level as a research paper.
Response: We thank the reviewer for the comment. As highlighted in the list of inclusion criteria on pag. 4, the review includes some conference papers (specifically articles [25], [26], [27]). We have highlighted this at the end of section 2.1 (line 221-223)
- Question: General suggestion: when citing a paper in the text, address the article with the last name of the first author, rather than simply using “In [34], sEMG signals…”. This also applies to the tables.
Response: Thank you for the suggestion. We have changed the citations adding the last name of the first author.
- Question: I do not think it is important information to go into detail and list all the EMG equipment, their figures, and the distribution of articles that used that equipment (Figures 4 & 5).
Response: We thank the reviewer for the important comment. Since, as highlighted in the introduction, the advantage of sEMG technology over classical ones is its greater diffusion, portability, cost, and usability, we feel it is appropriate to highlight the many commercial solutions currently available, which highlight a growing market. It is also important, of course, to understand their diffusion and distribution among solutions used by the scientific community (also to understand whether a specific technology platform marketed by a seller is difficult to use in reality)
- Question: Tables 3, 4 & 5 provided the same information as the corresponding paragraph. I suggest the authors rethink what information to put in the text and the tables and avoid redundancy.
Also, the paragraphs in the Results section mainly describe each individual paper, but I think if the authors can provide a more comprehensive summary and analysis of all the papers, it will be more beneficial to the readers.
Response: Thank you for your suggestion. In the revised version of the article, the marked redundant tables have been removed.
- Question: It is unclear why statistical analysis and artificial intelligence were put together in Figure 6 for comparison.
Response: Thank you for the important comment. In the modified version, the information about the used ‘classification’ techniques has been split into two specific figures.
- Question: The Discussion section mainly describes the findings again, but the summary/major findings of this review paper were clearly stated. From this review, there are many studies attempted to use EMG for sarcopenia detection and diagnosis, a clear statement of what has been found and EMG can or cannot do for this application needs to be thoroughly and clearly add.
Response: Thank you. Following your suggestion, the Discussion section has been revised and modified.
Reviewer 2 Report
Comments and Suggestions for Authors
This paper focuses on the application of surface electromyography (sEMG) in the detection and assessment of sarcopenia in the elderly, and explores related muscles, signal processing methods, feature extraction, and AI methods. Overall, the manuscript is well-organized and presented, but there are still some issues that need to be improved.
Main Comments:
- The manuscriptemphasizes the potential advantages of sEMG. Most of the research is based on laboratory environments, and it is necessary to supplement the verification results of sEMG technology in real clinical scenarios (such as community screening, long-term monitoring).
- The manuscriptintroduces signal processing techniques in detail, but most of the content is based on literature reviews, lacking actual data or case analysis. If possible, an experimental case comparing different signal processing methods could be provided to verify the effectiveness of certain methods in detecting sarcopenia.
- Thisarticle discusses various AI methods (SVM, RF, MLP, etc.), but lacks a discussion on the interpretability and clinical feasibility of these methods. It is recommended to supplement: which AI models perform best in sEMG data analysis? How to improve the interpretability of AI models to make them more suitable for medical applications?
- Thisarticle mainly refers to the EWGSOP2 and AWGS guidelines. However, are there other diagnostic criteria (such as North American standards)? Are there any studies comparing the impact of different diagnostic criteria on the application of sEMG? Is it possible to propose a new data-driven diagnostic criterion combined with sEMG signal characteristics?
Minor Comments:
- The manuscriptmentions multiple muscles (such as RF, BF, VL, etc.) and movement tasks, but the use of terms is not consistent enough. Some places use full names, and some use abbreviations. It is recommended to mark the full name when the term first appears and maintain consistency in subsequent use to improve readability.
- The manuscriptmentions the Butterworth band - pass filter (10–500 Hz), but does not explain the basis for choosing the high - frequency cut - off frequency (such as 500 Hz) (is it matched with the spectral characteristics of muscle electrical signals?).
To sum up, the reviewer hoped that the above comments would be helpful to the authors to revise their manuscript to make it more useful for the readers. It is recommended to supplement the discussion on the value of sEMG in clinical practice, data standardization, and the interpretability of AI. At the same time, optimize the use of some charts and terms to improve the readability and influence of the study.
Author Response
We thank the reviewer for the received feedbacks.
- Question: The manuscript emphasizes the potential advantages of sEMG. Most of the research is based on laboratory environments, and it is necessary to supplement the verification results of sEMG technology in real clinical scenarios (such as community screening, long-term monitoring).
Response: We thank the reviewer for the important comment. The technology examined in this review is not easily used in healthcare facilities for certification reasons. In the text, we have highlighted studies that operate in real-life scenarios (such as [38] line 691).
- Question: The manuscript introduces signal processing techniques in detail, but most of the content is based on literature reviews, lacking actual data or case analysis. If possible, an experimental case comparing different signal processing methods could be provided to verify the effectiveness of certain methods in detecting sarcopenia.
Response: Thank you for your comment. The aim of this review is not to evaluate the greater effectiveness of one signal processing technique over another in detecting sarcopenia, but to provide the reader with an understanding of the current state of the art regarding the use of commercial and non-commercial platforms for the detection of sarcopenia, highlighting which muscles are most involved, which signal processing and classification techniques are most widely used, and which devices are most tested by researchers.
- Question: This article discusses various AI methods (SVM, RF, MLP, etc.), but lacks a discussion on the interpretability and clinical feasibility of these methods. It is recommended to supplement: which AI models perform best in sEMG data analysis? How to improve the interpretability of AI models to make them more suitable for medical applications?
Response: Thank you for the suggestion. A paragraph (6.2) entitled “Interpretability and clinical feasibility of AI models” has been added in the revised version.
- Question: This article mainly refers to the EWGSOP2 and AWGS guidelines. However, are there other diagnostic criteria (such as North American standards)?
Response: We thank the reviewer for the comment. Other guidelines have been added, such as the International Working Group on Sarcopenia (IWGS), and the Foundation of National Institutes of Health (FNIH) (lines 48-49).
- Question: Are there any studies comparing the impact of different diagnostic criteria on the application of sEMG? Is it possible to propose a new data-driven diagnostic criterion combined with sEMG signal characteristics?
Response: Thank you for your observation. To the best of our knowledge, there is no published work comparing different diagnostic criteria on the same application.
- Question: The manuscript mentions multiple muscles (such as RF, BF, VL, etc.) and movement tasks, but the use of terms is not consistent enough. Some places use full names, and some use abbreviations. It is recommended to mark the full name when the term first appears and maintain consistency in subsequent use to improve readability.
Response: Thank you for your comment, the article has been revised according to your suggestion.
- Question: The manuscript mentions the Butterworth band - pass filter (10–500 Hz), but does not explain the basis for choosing the high - frequency cut - off frequency (such as 500 Hz) (is it matched with the spectral characteristics of muscle electrical signals?).
Response: Thank you for your comment. Applying a band-pass filter between 20 and 450 Hz is an effective strategy in electromyography (EMG) to minimize noise while preserving essential information about muscle activity. This frequency range includes the primary components of the EMG signal, which are typically found between 10 and 500 Hz. Filtering frequencies below 20 Hz reduces motion artefacts and baseline noise, while filtering above 450 Hz helps to eliminate high frequency noise such as electromagnetic interference.
We have included this information in the text (line 506-510) and added a reference:
[70] De Luca, C. J., Gilmore, L. D., Kuznetsov, M., & Roy, S. H. (2010). Surface EMG signal filtering: Motion artefact and baseline noise contamination. Journal of Biomechanics, 43(8), 1573-1579.
Reviewer 3 Report
Comments and Suggestions for Authors
Dear authors,
1. If you used an extended version of the Preferred Reporting Items for Systematic Reviews and Meta-Analyses (PRISMA) [21] was used as the methodological framework for this review, you should mention it in the Title. If you have mentioned [21] reference: Moher, David, et al. "Preferred reporting items for systematic reviews and meta-analyses: the PRISMA statement." International journal of surgery 8.5 (2010): 336-341.
Please follow the guidelines that you refer to for developing the Systematic review and Meta-Analyses (PRISMA) checklist, you can find it here: https://www.prisma-statement.org/.
2. In the introduction part formulate the aim of the study, according to PRISMA, and the introduction lacks emphasis on why systematic review and metanalysis are needed.
3. PubMed, and ELSEVIER Scopus are narrow databases for the study, involving Signal Processing Techniques, Significant Features, and Artificial Intelligence Approaches. Please widen the sources of the information. E.g. IEEE, AMC Digital Library, etc.
The English could be improved to express the research more clearly. E.g. Inclusion and exclusion criteria should be rewritten more formally.
Author Response
We thank the reviewer for the received feedbacks.
Question: If you used an extended version of the Preferred Reporting Items for Systematic Reviews and Meta-Analyses (PRISMA) [21] was used as the methodological framework for this review, you should mention it in the Title. If you have mentioned [21] reference: Moher, David, et al. "Preferred reporting items for systematic reviews and meta-analyses: the PRISMA statement." International journal of surgery 8.5 (2010): 336-341.
Please follow the guidelines that you refer to for developing the Systematic review and Meta-Analyses (PRISMA) checklist, you can find it here: https://www.prisma-statement.org/.
Response: Thank you for the suggestion. We have changed the title to “A Systematic Review of Surface Electromyography in Sarcopenia: Muscles Involved, Signal Processing Techniques, Significant Features, and Artificial Intelligence Approaches.” Also, we have checked the PRISMA guidelines and modified the following sections: Abstract (lines 6-21); 2.1 Article Selection, Inclusion and Exclusion Criteria (lines 216-219), and 8. Discussion according to the PRISMA Checklist.
- Question: In the introduction part formulate the aim of the study, according to PRISMA, and the introduction lacks emphasis on why systematic review and metanalysis are needed.
Response: Thank you for the comment. We formulated the purpose of the review according to PRISMA and emphasized why a systematic review is needed in the Introduction (lines 110-126).
- Question: PubMed, and ELSEVIER Scopus are narrow databases for the study, involving Signal Processing Techniques, Significant Features, and Artificial Intelligence Approaches. Please widen the sources of the information. E.g. IEEE, AMC Digital Library, etc.
Response: We thank the reviewer for the suggestion. We have added IEEE as a source, the used query has been added to Table 1, and the PRISMA flow diagram has been modified (Figure 2.).
- Comment: The English could be improved to express the research more clearly. E.g. Inclusion and exclusion criteria should be rewritten more formally.
Response: Thanks for the comment. English has been revised, and the inclusion and exclusion criteria have been rewritten more formally (lines 168-185).
Round 2
Reviewer 3 Report
Comments and Suggestions for Authors
Dear Authors,
Thank you for your revised submission. I appreciate the improvements made to the manuscript and recognize that you have addressed the concerns raised in the review process. Your adherence to the PRISMA guidelines has enhanced the methodological rigor and clarity of the systematic review.
I find that the revisions have strengthened the manuscript, particularly in the structured synthesis of literature and the discussion of methodological heterogeneity. The improved focus on standardized guidelines and future research directions adds valuable insights.
Overall, I believe the manuscript is now well-structured and contributes meaningfully to the field. I appreciate your efforts in refining the work.
Author Response
We are glad to have met the reviewer's requirements.